# National trends in emergency department closures, mergers, and utilization, 2005-2015

**Arjun K. Venkatesh**[1,2☯]*, **Alexander Janke**[1☯], **Craig Rothenberg**[1☯], **Edwin Chan**[1☯], **Robert D. Becher**[3☯]

**1** Department of Emergency Medicine, Yale University School of Medicine, New Haven, Connecticut, United States of America, **2** Center for Outcomes Research and Evaluation, Yale University School of Medicine, New Haven, Connecticut, United States of America, **3** Section of General Surgery, Trauma & Surgical Critical Care, Yale University School of Medicine, New Haven, Connecticut, United States of America

☯ These authors contributed equally to this work.
\* arjun.venkatesh@yale.edu

## Abstract

### Study objectives

To describe nationwide hospital-based emergency department (ED) closures and mergers, as well as the utilization of emergency departments and inpatient beds, over time and across varying geographic areas in the United States.

### Methods

Observational analysis of the American Hospital Association (AHA) Annual Survey from 2005 to 2015. Primary outcomes were hospital-based ED closure and merger. Secondary outcomes were yearly ED visits per hospital-based ED and yearly hospital admissions per hospital bed.

### Results

The total number of hospital-based EDs decreased from 4,500 in 2005 to 4,460 in 2015, with 200 closures, 138 mergers, and 160 new hospital-based EDs. While yearly ED visits per hospital-based ED exhibited a 28.6% relative increase (from 25,083 to 32,248), yearly hospital admissions per hospital bed had a 3.3% relative increase (from 45.4 to 43.9) from 2005 to 2015. The number of hospital admissions and hospital beds did not change significantly in urban areas and declined in rural areas. ED visits grew more uniformly across urban and rural areas.

### Conclusions

The number of hospital-based ED closures is small when accounting for mergers, but occurs as many more patients are presenting to a stable number of EDs in larger health systems, though rural areas may differentially affected. EDs were managing accelerating patient volumes alongside stagnant inpatient bed capacity.

**Data Availability Statement:** Data is available for a fee at the following url: https://www.ahasurvey.org/taker/asindex.do. Data can be obtained by searching for "AHA Annual Survey Database." The

authors confirm that they did not have special access to the data that other researchers would not have.

**Funding:** Dr. Venkatesh reports support from the Emergency Medicine Foundation Health Policy Scholar Award (https://www.emfoundation.org/) and the Yale Center for Clinical Investigation KL2TR000140 from the National Center for Advancing Translational Science (https://ncats.nih.gov/), a component of the National Institutes of Health (NIH), as well as support by the National Academy of Medicine -American Board of Emergency Medicine Anniversary Fellowship (https://nam.edu/). The funders had no role in study design, data collection and analysis, decision to publish, or preparation of the manuscript.

**Competing interests:** The authors have declared that no competing interests exist.

# Introduction

## Background and importance

U.S. acute and hospital-based care capacity have evolved over time, and their relevance is all the more salient given the challenges of the COVID-19 pandemic. This evolution occurs in the context of recent hospital and health system consolidation, with larger facilities capturing greater market share and consolidating care among fewer, more integrated systems nationwide [1]. Consolidation had been driven by numerous economic forces and uncertainty in the political climate following passage of the Affordable Care Act [2,3]. News reports have suggested that amidst this wave of consolidation hospitals and their associated emergency departments (EDs) may often be closed even in areas where readily-available alternative care sites are not present [4], though there is little empirical investigation to support this claim. Prior work has documented several impacts of consolidation including loss of market competition [5–7], incentivized coordination of care [8], and in select geographies, reduced access to hospital-based emergency care and poorer outcomes [9].

Access to care remains a challenge in the U.S., and this challenge is more than simply a question of having health insurance. Access is a complex multidimensional construct inclusive of coverage, a source for healthcare services, timely availability of services and a capable workforce to deliver services [10]. Although the Emergency Medical Treatment and Labor Act (EMTALA) protects patients' access to emergency care-in that all who present must receive a medical screening exam, meaningful access to timely hospital and emergency care services is primarily driven by geography, or simply put the presence of a hospital-based emergency department close by [11].

Prior work based on national survey data found that between 1990 and 2009, the number of hospital-based EDs declined by 27% in non-rural areas, and for-profit owned EDs, those in competitive markets and those with safety-net status were more likely to close [12]. In the decade since, the financial underpinnings of emergency and hospital care in the U.S. has changed substantially as a result of growing profits from hospital-based ambulatory diagnostic and treatment centers as well as expansion of both Medicaid and Affordable Care Act-based insurance exchange populations [13]. Amidst a wave of consolidation, some hospital-based EDs may remain open as mergers favor health system network expansion rather than closure of a single facility [14].

Timely access to hospitals and EDs has been associated with outcomes after trauma and myocardial infarction, as well as inpatient mortality [9,15,16]. Both hospital and ED closures have been described as limiting access and associated with worse clinical outcomes at the state level in cardiovascular and trauma care [9,16]. Furthermore, for EDs that remain open, ED visit volumes have steadily increased year-over-year [17], and this may be one contributor to ED crowding and its downstream consequences [18,19]. Despite the importance of timely access to EDs with appropriate capacity, there has been no recent national assessment of ED closures and mergers as well as hospital capacity that accounts for both recent health system consolidation and the passage of the Affordable Care Act in 2010. The pressures placed on health systems, and the likelihood of ongoing surges in the Covid-19 outbreak, make the capacity of EDs and hospitals all the more salient.

## Goals of this investigation

The goal of this investigation was to describe hospital-based ED closures and mergers in the period from 2005–2015. We characterized closures and mergers by urban-rural designation, ED visit volumes, as well as facility teaching and for-profit status. Our secondary objective was

to characterize yearly ED visits per hospital-based ED and yearly hospital admissions per hospital bed over time and across urban-rural designation.

## Methods

### Study design and setting

This was a secondary analysis of the American Hospital Association (AHA) Annual Survey Database with data from years 2005 through 2015 [20]. The Annual Survey Database is a proprietary dataset maintained by the AHA compiling yearly results of an AHA-directed nationwide survey of all hospitals in the United States (including the 50 states, and excluding Puerto Rico, Mariana Islands, Marshall Islands, Virgin Islands, Guam, American Samoa) as well as data from the U.S. Census Bureau and hospital accrediting bodies. Response rates among hospitals ranged from 84% to 92% in the years analyzed. We specifically excluded Veterans Affairs and other hospitals related to the federal government/military (government, nonfederal facilities were included) as defined by control code in the AHA database (excepting the Indian Health Service, which was included). Our analysis included all short-term acute care hospitals providing general medical and surgical services, as defined by service code, that are listed in at least two contiguous years of data. Stand-alone pediatric facilities are identified by a different service code and therefore excluded from the present analysis. **S1 Fig** gives counts for included hospitals by exclusion criteria.

### Measurements

Hospitals were linked year-over-year, and we defined a hospital-based ED for those hospitals that reported at least two consecutive years with >500 ED visits per year. This approach, requiring 2 years of consecutive data and >500 ED visits per year, was set to exclude potentially spurious sites in this dataset composed of aggregated hospital survey responses. As a point of reference, there was an estimated 28.4 million ED visits and 1,855 rural EDs in 2016, so that the average yearly ED visits per rural ED would be 15,309, while sites with <500 ED visits per year were felt to be less likely to represent a bona fide point of emergency care services. AHA data were combined with American Community Survey data [21] from the U.S. Census Bureau at the zip-code level to characterize population level characteristics as they are related to proximity to an ED, as well as Rural Urban Commuting Area (RUCA) code, a scheme for delineating urban and rural areas based on the American Community Survey. We defined four categories (designated according to recommendation of the Rural Health Research Center): urban (RUCA codes 1.0, 1.1, 2.0, 2.1, 3.0, 4.1, 5.1, 7.1, 8.1, 10.1), large rural (4.0, 4.2, 5.0, 5.2, 6.0, 6.1), small rural (7.0, 7.2, 7.3, 7.4, 8.0, 8.2, 8.3, 8.4, 9.0, 9.1, 9.2), and isolated (10.0, 10.2, 10.3, 10.4, 10.5, 10.6) [22]. These groupings are standardized from the Rural Health Research Center, and sort tracts according to patterns of commute (for example, a rural tract with 30–50% of commuting secondary flow to an urban area is categorized as urban) [23]. We defined ED visit volume categories for analysis consistent with the Emergency Department Benchmarking Alliance and the American College of Emergency Physicians Clinical Emergency Data Registry (500 to 20k, 20-40k, 40-60k, 60-80k, 80k+) [24]. Hospital-based EDs were characterized by each of ED visit volume categories, urban-rural designation, teaching status, and for-profit status.

### Outcomes

Our primary outcomes were the number of hospital-based ED closures and hospital-based ED mergers (sites that remained open but were subject of a merger with another health system) by

year. Available with each year of the AHA dataset, there is a supplemental document with a described reason for each hospital site removed or added to the data. A hospital-based ED closed if the site no longer reported >500 ED visits in a given year, or if the hospital closed. By defining closures with respect to ED visit counts in two contiguous years from available data in 2005 through 2015, closures were defined in the analysis only for the period from 2006–2014. If a site was removed from the dataset and their listed description was not for a merger, but some other change, these were independently reviewed by research staff to clarify if the site had indeed closed. A hospital-based ED was considered to have closed if the site was not reported in a subsequent year, even with a listed reason for removal as merger, if that geographic site was not found to be operating a hospital-based ED site in 2015 after independent review by research staff. Some hospitals may remain open while their associated ED is closed, and these fit our outcome definition for a hospital-based ED closure. Operation of ED sites at the specified locations were primarily assessed via published notes or histories from the associated healthcare system. When such information was not available, ongoing operation was assessed via a combination of calls to the appropriate healthcare system and internet searches for the appropriate locations. Any site deemed permanently closed was confirmed by at least two separate sources. If the hospital-based ED was in fact still in operation in 2015 as clarified by external sources, but removed from the dataset for a listed merger, then the most recently available year of data in AHA was copied forward to 2015. These sites are defined as merged for the purposes of our analysis (that is, the reported outcome for merger is a site that remained in operation). This approach ensured that an ED was not defined as closed when in fact the site remained in operation and simply consolidated with a larger system. Total ED visits rose year-over-year for nearly all sites (>90%), which suggests that this approach of copying data forward where sites were removed from the dataset but in fact remained open as part of a hospital or health system merger would provide a conservative estimate of total ED visit volume. Those sites that were listed as new in the dataset but 'not previously registered' were not listed as new hospitals. We include those listed as 'new' or 'newly re-opened' as opening facilities, with the caveat that these facilities must meet the same criteria for hospital-based ED as above.

We tabulated total ED visit volumes, total hospital-based EDs, total acute care admissions, and total inpatient beds by year. To summarize these changes over time, we then calculated the nationwide average yearly ED visits per hospital-based ED and the nationwide average yearly hospital admissions per hospital bed.

## Analysis

For the primary analysis, year-over-year trends in hospital-based ED closures and mergers were characterized by urban-rural location, ED visit volumes, as well as facility teaching and for-profit status. Univariable and multivariable logistic regression models were constructed to estimate the likelihood of hospital-based ED closure and merger among the following covariates: urban-rural location, ED visit volume, facility teaching and for-profit status. Each observation for model estimation is a unique hospital-based ED, and we assessed for multicollinearity amongst independent variables by obtaining associated variance inflation factors. Both unadjusted and adjusted odds ratios and 95% confidence intervals were calculated for the relationship between urban-rural designation, ED visit volume, teaching status, as well as for-profit status and hospital-based ED closure and merger during the study period. The geographic location of closures and mergers was represented according to the hospital referral region (HRR) where the closure or merger took place, subject to the limitation that only those HRRs with 5 or greater hospital-based EDs were depicted to protect hospital

identities according to AHA data use agreement. This was to appropriately limit the granularity of the results according the AHA's terms of data use. We furthermore describe the year-over-year trends in ED visit and acute care hospitalization volumes according to urban-rural designation. Our analysis does not include time trends, as event rates for closure and merger were fairly uncommon to derive stable estimates, and variation in other ED characteristics generally static over time, to provide a definitive analysis of how factors associated with closure and merger have changed over time. The secondary analysis consisted of tabulating number of ED visits, number of hospital-based EDs, number of inpatient stays, and number of available hospital beds by year. The outcomes were then defined as the ratio of ED visits to EDs and hospital admissions to hospital beds. The percent change from 2005 baseline through 2015 in these two outcomes was calculated, stratified by urban-rural designation (urban, large rural, small rural, isolated). **S1 Table** provides the STROBE checklist for the present investigation. All analyses were performed in R (R Foundation, version 4.0.1).

## Results

The total number of hospital-based EDs listed fell from 4,500 in 2005 to 4,460 in 2015, which includes the addition of 160 new ED facilities. Overall, there were 200 hospital-based ED closures identified and 138 hospital-based ED mergers (where sites remained open but were subject to a health system merger). As a point of comparison, our approach identified 4,460 EDs in 2015, similar to the 4,545 EDs in the same year according to data from the Healthcare Cost and Utilization Project [25]. In the final year of analysis, there were 2,296 urban hospital-based EDs and 2,164 large rural, small rural, or isolated hospital-based EDs. The event rates for closure and merger, as a percent of overall EDs, remained <1% across subgroups of EDs and across years.

Tables **1** and **2** depict characteristics and risk factors for closures and mergers. We find that closures and mergers were more likely for smaller EDs and those in urban areas. **Fig 1** depicts maps of the United States for the location of ED closures and mergers in the study period within hospital referral region, respectively [26]. These demonstrate that closures are relatively concentrated in the south and notably Texas, while mergers are more common in the Northeast, Ohio, and Texas.

In our secondary analysis of average yearly ED visits and hospital admissions, nationwide ED visits are increasing, while inpatient hospitalizations have been relatively stable and the total number of inpatient beds has actually slightly declined (**Table 3**). These trends result in an increase in average yearly ED visits per hospital-based ED, while inpatient hospitalizations per hospital bed is largely unchanged in the period from 2005 to 2015 as in **Fig 2**. Across all urban-rural categories, ED visits have made a divergence from inpatient hospitalizations. In non-urban areas, though, inpatient hospitalizations have decreased (>10% decline in each of large rural, small rural, and isolated) more so than in urban areas (<5% decline). **Fig 3** depicts these trends by urban-rural designation, demonstrating growth of ED visits in each urban-rural categories with relatively less change in hospital admissions and hospital beds.

### Limitations

First, hospital reporting in the AHA dataset is not entirely standardized, and the dataset relies on self-report from a representative of each individual hospital responding to the survey. This introduces the risk of inaccurate reporting or missing data. Second, though we regard this also as a potential strength of our analysis, hospital mergers may also represent a limitation. It is possible that what appear as ED closures in our dataset are in fact simple hospital consolidation into larger systems with ED reporting absorbed in to the larger hospital. Our analysis

**Table 1. Characteristics and risk factors for hospital-based ED closures, 2006–2014.**

| | 2005 | 2006 | 2007 | 2008 | 2009 | 2010 | 2011 | 2012 | 2013 | 2014 | OR [95% CI] | aOR [95% CI] |
|---|---|---|---|---|---|---|---|---|---|---|---|---|
| Total Hospital-Based EDs | 4,500 | 4,529 | 4,541 | 4,538 | 4,535 | 4,547 | 4,552 | 4,546 | 4,551 | 4,505 | | |
| Total Closures | --- | 27 | 17 | 23 | 17 | 10 | 31 | 15 | 31 | 29 | --- | --- |
| Urbanicity | | | | | | | | | | | | |
| Urban | --- | 24 | 13 | 16 | 11 | 8 | 17 | 6 | 20 | 6 | RG | RG |
| Large Rural | --- | 0 | 4 | 2 | 4 | 0 | 3 | 3 | 2 | 5 | 0.61 [0.38–0.95] | 0.39 [0.23–0.63] |
| Small Rural | --- | 0 | 0 | 2 | 0 | 2 | 7 | 1 | 4 | 12 | 0.60 [0.39–0.91] | 0.27 [0.16–0.45] |
| Isolated | --- | 3 | 0 | 3 | 2 | 0 | 4 | 5 | 5 | 6 | 1.03 [0.66–1.55] | 0.51 [0.3–0.84] |
| ED Visit Volume | | | | | | | | | | | | |
| 500 to 20,000 | --- | 14 | 7 | 12 | 14 | 5 | 21 | 12 | 27 | 28 | 13.76 [3.06–242.61] | 21.16 [4.22–387.53] |
| 20,000–40,000 | --- | 9 | 9 | 10 | 3 | 5 | 6 | 2 | 3 | 1 | 10.56 [2.30–187.30] | 11.89 [2.39–217.13] |
| 40,000–60,000 | --- | 3 | 0 | 0 | 0 | 0 | 3 | 1 | 1 | 0 | 3.06 [0.55–56.92] | 3.1 [0.54–58.56] |
| 60,000–80,000 | --- | 1 | 0 | 1 | 0 | 0 | 1 | 0 | 0 | 0 | 3.06 [0.45–60.21] | 3 [0.43–59.39] |
| 80,000+ | --- | 0 | 1 | 0 | 0 | 0 | 0 | 0 | 0 | 0 | RG | RG |
| Teaching Status | | | | | | | | | | | | |
| Teaching | --- | 1 | 2 | 0 | 0 | 0 | 1 | 0 | 0 | 0 | 0.38 [0.13–0.84] | 1 [0.29–2.67] |
| Non-Teaching | --- | 26 | 15 | 23 | 17 | 10 | 30 | 15 | 31 | 29 | RG | RG |
| Profit Status | | | | | | | | | | | | |
| Not-For-Profit | --- | 10 | 11 | 14 | 11 | 6 | 10 | 7 | 11 | 16 | RG | RG |
| For-Profit | --- | 12 | 5 | 9 | 2 | 3 | 13 | 5 | 16 | 11 | 2.49 [1.79–3.43] | 1.92 [1.34–2.74] |
| Government* | --- | 3 | 0 | 0 | 0 | 0 | 0 | 0 | 0 | 0 | --- | |

Results are nationwide inventory of hospital-based EDs from American Hospital Association Annual Survey data. This only includes results from the last year of available data before the site was removed from the dataset for closure. As closures are defined by 2 consecutive years with fewer than 500 ED visits, closures are not defined in the first and last year of available data (2005 and 2015). Odds ratios were estimated using a multivariable logistic regression model.

*Government category excluded from adjusted odds ratio calculations due to limited sample size. ED = emergency department, RG = reference group, aOR = Adjusted odds ratio, CI = confidence interval.

attempted to specifically address this issue, as each site dropped from the dataset was individually reviewed. Third, an important aspect of the Affordable Care Act was the initiation of statewide Medicaid expansions in 2014, but because closure and merger are defined by two consecutive years of data for a given hospital-based ED, we do not have results after the expansions. Finally, though our analysis can address the distribution of EDs across urban and rural areas, it does not address more specific care processes such as time-to-arrival for ED evaluation, or the change in how far a person may need to travel to the next closest ED when one closes, that may have changed in the period we analyzed. There is a great deal of literature on the topic of care for time-sensitive conditions, and future analyses should explore the relationship between the trends observed in this work and specific patient outcomes for time-sensitive illness [9,15,16].

## Discussion

The overall number of hospital-based EDs in the United States changed very little from 2005 to 2015. Furthermore, mergers across hospitals and health systems appear to occur with similar frequency as closures nationwide. This is part of an overall trend in American healthcare towards organizational consolidation, with potential implications for one of the key aspects of access to care: geographic availability. This may be attenuated by the Affordable Care Act in 2010, and subsequent to statewide Medicaid expansions and other shifts in underlying population insurance status beyond 2014, hospital-based EDs may be less prone to closure. Our

**Table 2. Characteristics and risk factors for hospital-based ED mergers, 2005 to 2014.**

| | 2005 | 2006 | 2007 | 2008 | 2009 | 2010 | 2011 | 2012 | 2013 | 2014 | OR [95% CI] | aOR [95% CI] |
|---|---|---|---|---|---|---|---|---|---|---|---|---|
| Total Hospital-Based EDs | 4,500 | 4,529 | 4,541 | 4,538 | 4,535 | 4,547 | 4,552 | 4,546 | 4,551 | 4,505 | | |
| Total Mergers | 18 | 10 | 9 | 10 | 22 | 13 | 23 | 20 | 19 | 12 | --- | --- |
| Urbanicity | | | | | | | | | | | | |
| Urban | 16 | 9 | 8 | 8 | 20 | 12 | 23 | 15 | 15 | 9 | RG | RG |
| Large Rural | 2 | 0 | 1 | 1 | 2 | 1 | 0 | 1 | 0 | 2 | 0.23 [0.11–0.41] | 0.22 [0.11–0.42] |
| Small Rural | 0 | 0 | 0 | 1 | 0 | 0 | 0 | 2 | 4 | 1 | 0.15 [0.07–0.28] | 0.14 [0.06–0.3] |
| Isolated | 0 | 1 | 0 | 0 | 0 | 0 | 0 | 2 | 0 | 0 | 0.09 [0.02–0.24] | 0.09 [0.02–0.26] |
| ED Visit Volume | | | | | | | | | | | | |
| 500 to 20,000 | 7 | 4 | 2 | 6 | 8 | 5 | 10 | 6 | 13 | 3 | 2.93 [0.91–17.92] | 5.06 [1.47–32] |
| 20,000–40,000 | 7 | 4 | 3 | 2 | 8 | 7 | 4 | 7 | 4 | 1 | 6.23 [1.93–38.18] | 4.83 [1.43–30.28] |
| 40,000–60,000 | 3 | 2 | 4 | 1 | 4 | 0 | 6 | 4 | 2 | 5 | 4.69 [1.38–29.34] | 3.74 [1.07–23.7] |
| 60,000–80,000 | 1 | 0 | 0 | 1 | 4 | 1 | 2 | 0 | 0 | 3 | 3.89 [1.01–25.47] | 3.31 [0.85–21.8] |
| 80,000+ | 0 | 0 | 0 | 0 | 2 | 0 | 1 | 3 | 0 | 0 | RG | RG |
| Teaching Status | | | | | | | | | | | | |
| Teaching | 0 | 0 | 0 | 0 | 0 | 0 | 2 | 1 | 0 | 0 | 0.59 [0.23–1.23] | 0.43 [0.13–1.09] |
| Non-Teaching | 18 | 10 | 9 | 10 | 22 | 13 | 21 | 19 | 19 | 12 | RG | RG |
| Profit Status | | | | | | | | | | | | |
| Not-For-Profit | 8 | 2 | 6 | 8 | 9 | 6 | 13 | 15 | 8 | 8 | RG | RG |
| For-Profit | 8 | 6 | 2 | 1 | 11 | 6 | 9 | 1 | 7 | 2 | 2.73 [1.89–3.88] | 1.79 [1.2–2.63] |
| Government* | 0 | 0 | 0 | 0 | 0 | 0 | 0 | 0 | 0 | 0 | --- | --- |

Results are nationwide inventory of hospital-based EDs from American Hospital Association Annual Survey data. This only includes results from the last year of available data before the site was removed from the dataset for merger. As mergers were defined for hospitals that listed merger as reason for removal from the dataset but were open in the following year, mergers are not defined in 2015. Odds ratios were estimated using a multivariable logistic regression model.

*Government category excluded from adjusted odds ratio calculations due to limited sample size. ED = emergency department, RG = reference group, aOR = Adjusted odds ratio, CI = confidence interval.

findings suggest that, throughout the full analysis period, hospital-based EDs most likely to be closed or merged are low-volume facilities in urban areas; however given the reliance of rural communities in which only a single ED may be available, rural ED closures may carry greater impact on population access to emergency care. While benefits may accrue from system

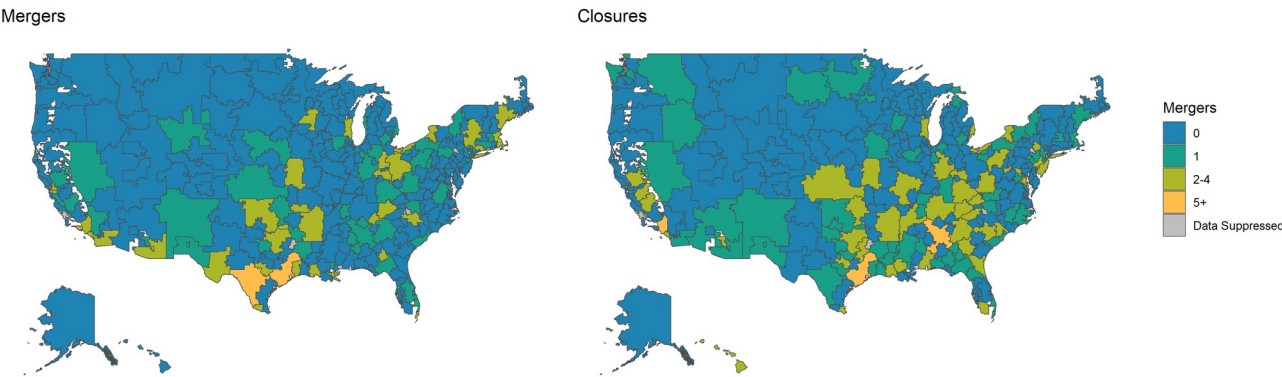

**Fig 1. Hospital-based ED closures and mergers nationwide: 2005–2014.** Source/Notes: Results are nationwide hospital-based ED closures and mergers nationwide from 2005–2014 from the American Hospital Association Annual Survey data. Shaded areas represent hospital referral region. Some data is suppressed according to terms of American Hospital Association data use agreement to protect hospital identity.

**Table 3. Nationwide ED and inpatient utilization, 2005–2015.**

|  | 2005 | 2006 | 2007 | 2008 | 2009 | 2010 | 2011 | 2012 | 2013 | 2014 | 2015 |
|---|---|---|---|---|---|---|---|---|---|---|---|
| Total Hospital-Based EDs | 4,500 | 4,529 | 4,541 | 4,538 | 4,535 | 4,547 | 4,552 | 4,546 | 4,551 | 4,505 | 4,460 |
| ED Visits (Millions) | 113 | 117 | 120 | 122 | 126 | 127 | 130 | 134 | 135 | 138 | 144 |
| Average ED Visits Per ED | 25,083 | 25,775 | 26,336 | 26,837 | 27,819 | 27,946 | 28,488 | 29,404 | 29,638 | 30,641 | 32,248 |
| Urban | 38,446 | 39,723 | 40,786 | 41,370 | 42,972 | 43,321 | 44,308 | 45,889 | 46,174 | 48,116 | 50,385 |
| Large Rural | 20,565 | 21,067 | 21,387 | 21,926 | 22,444 | 22,155 | 22,572 | 22,607 | 23,020 | 23,477 | 24,617 |
| Small Rural | 8,295 | 8,361 | 8,534 | 8,742 | 8,823 | 8,683 | 8,727 | 8,842 | 8,796 | 8,736 | 9,019 |
| Isolated | 4,572 | 4,576 | 4,665 | 4,703 | 4,775 | 4,866 | 4,601 | 4,825 | 4,673 | 4,787 | 5,280 |
| Inpatient Beds | 750,869 | 765,447 | 769,434 | 769,458 | 767,475 | 770,238 | 764,409 | 764,733 | 764,228 | 757,622 | 757,374 |
| Admits (Thousands) | 34,084 | 34,395 | 34,585 | 34,817 | 34,658 | 34,491 | 34,236 | 33,957 | 33,379 | 32,881 | 33,230 |
| Average Admits Per Bed | 45.4 | 44.9 | 44.9 | 45.2 | 45.2 | 44.8 | 44.8 | 44.4 | 43.7 | 43.4 | 43.9 |
| Urban | 48.8 | 48.3 | 48.3 | 48.6 | 48.6 | 48.1 | 48.1 | 47.8 | 47.1 | 46.8 | 47.2 |
| Large Rural | 40.8 | 40.1 | 39.9 | 40.0 | 39.3 | 38.9 | 38.4 | 37.4 | 36.5 | 36.0 | 36.7 |
| Small Rural | 26.9 | 26.6 | 26.2 | 26.2 | 26.0 | 25.6 | 25.5 | 24.3 | 23.6 | 22.7 | 22.6 |
| Isolated | 18.7 | 18.6 | 18.5 | 18.6 | 17.9 | 17.6 | 17.4 | 16.5 | 16.2 | 15.5 | 15.8 |

Results are nationwide inventory of emergency department (ED) and inpatient availability and utilization from American Hospital Association Annual Survey data. Beds include all reported general medical/surgical hospital beds reported by each hospital in the sample.

consolidation, for example if it facilitates transitions of care between hospitals, information technology adoption and data sharing, and better specialization for specific conditions, if capable providers and systems are not open and accessible, patients may suffer.

Our results demonstrate far fewer ED closures overall than found by earlier work using similar national survey data [12] that did not account for mergers. Notably, we individually curated data on mergers in our analysis that may have previously caused sites to be misinterpreted as closed. This discrepancy alone would change our outcome of hospital-based ED closures in the full study period by 166 events, representing a 78% error in the outcome reporting. Because health system consolidation has become so common, any national analysis of ED closures should account for this trend. This phenomena may also explain our finding that small EDs in urban areas are more likely to close, as opposed to rural sites, as they may be more likely to be subject to competitive pressures while small rural EDs that are more commonly threatened with closure in media reports may be able to raise political or community attention about the essential role of rural EDs in providing access to complex care not available

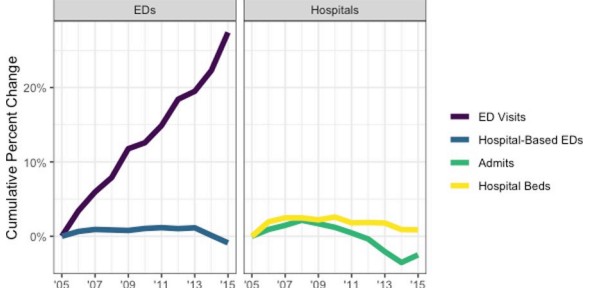

**Fig 2. Nationwide percent change from 2005 baseline in ED and inpatient utilization.** Source/Notes: Results are nationwide inventory of emergency department (ED) and inpatient availability and utilization from American Hospital Association Annual Survey data. Measures include total yearly ED visits, hospital-based EDs, total yearly hospital admissions, hospital beds.

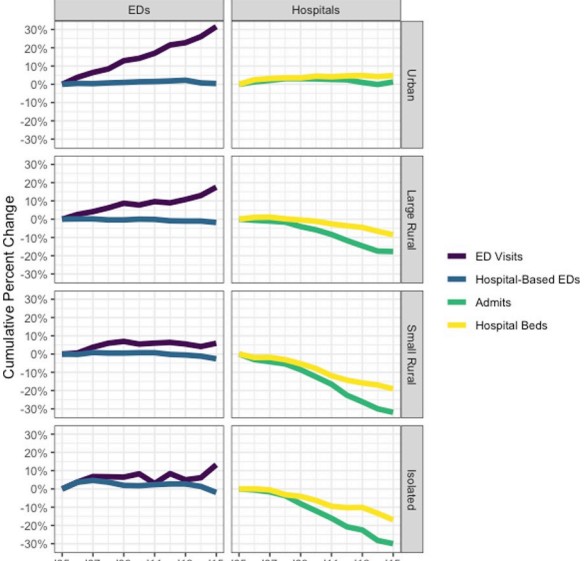

**Fig 3. Nationwide percent change from 2005 baseline in ED visits and inpatient hospitalizations by urban-rural category.** Source/Notes: Results are nationwide inventory of emergency department (ED) and inpatient availability and utilization from American Hospital Association Annual Survey data. Measures include total yearly ED visits, hospital-based EDs, total yearly hospital admissions, hospital beds.

elsewhere. Furthermore, our inclusion criteria are more stringent than in Hsia et al, as we require two consecutive years of response that include 500 or more yearly ED visits to be included as a hospital-based ED, as well as two consecutive years with ED visits to confirm closure. This means we may be avoiding bias created by imperfect response rates.

The geographic distribution of hospital-based ED closures in our analysis mirrors and extends upon work by Schuur et al. that inventoried free-standing EDs across the U.S. [27]. Specifically, we found higher absolute and relative hospital-based ED closures in states such as Florida, Ohio, and Texas that have experienced marked growth in free-standing emergency centers (**Fig 1**), and this may be an indicator of increased financial volatility of hospital-based ED care where free-standing EDs alter the competitive landscape. The degree to which competitive forces of multiple EDs may impact hospital access should be explored in future research, particular considering that free-standing centers may concentrate in areas with well-insured populations [28], while those the Medicaid or uninsured populations may concentrate near now-closing hospital-based EDs. Changes in propensity to utilize free-standing versus hospital-based emergency care may impact access to traditional hospital-based services such as coronary revascularization or advanced trauma care.

Consistent with federal statistics [25], our study shows a steady significant rise in ED visits nationwide and consistent across urban-rural designations. Though the stable number of hospital-based EDs is encouraging, ED visit growth occurs despite a number of forces that might otherwise decrease them. There has been an extraordinary rise of retail clinics, urgent care centers, telemedicine and alternative sites of care [29,30]. Simultaneous and persistently-increasing ED visit volumes may be contributing to the problem of ED crowding. More importantly though, we find inpatient capacity (as measured here by number of hospital beds) is not growing and this is an often-cited cause of crowding [31]. Limited hospital bed availability may exacerbate ED boarding and the known consequences of ED crowding including increased mortality [32]. Though urgent care and telemedicine offer alternatives to the ED for lower acuity conditions (those that do not require inpatient beds), these venues would not offset the

limitations of inpatient capacity if ED visit acuity has not changed in recent years. Indeed, there is evidence that patients discharged from our nation's EDs are becoming older and more comorbid each year [33]. It may be that EDs are better at the management of higher acuity presentations (obviating the need for admission).

Surprisingly, we found different trajectories for inpatient utilization intensity between urban and rural communities. Though both urban and rural areas are seeing greater ED utilization intensity, rural areas are uniquely impacted by large declines in inpatient utilization intensity (**Table 3**) driven by both decreased available beds and decreased admissions (**Fig 3**). Though this may be due to differences in the level of illness across population, those in smaller communities may increasingly rely on the ED for healthcare services access and be at increased risk of "under hospitalization" [34]. This may increase reliance on EDs as sites for acute, unscheduled ambulatory care in rural communities [35].

There may be numerous unintended consequences to our observed trends of accelerated ED visitation alongside decreasing inpatient utilization, including a trend towards EDs tolerating more risk and serving as substitutes to inpatient wards for extensive work ups. However, these trends may have a complementary mechanism as well: prior work has documented the evolving role of the hospital-based ED into an acute diagnostic center capable of providing access for unscheduled care unavailable in other outpatient settings [36] and applying technology to accelerate risk stratification and lower hospitalization rates [37]. Policymakers and health system leaders observing these trends should focus on accruing the benefits of the ED's expanding role while limiting collateral damage due to crowding or decreased geographic access.

Though hospital-based EDs are closing and consolidating, our findings suggest that there has been very little change in the overall number of facilities nationwide, with more patients presenting to a stable number of emergency departments in larger, more complex systems, though rural areas may be differentially affected. Emergency departments are managing increased visit volumes alongside stable inpatient bed capacity. This may suggest increased ED or system-level efficiency at evaluating and managing patients in the ED setting without subsequent hospital admission.

## Supporting information

**S1 Fig. Hospital analysis sample exclusions.** Source/Notes: Data from the American Hospital Association Annual Survey data.
(TIF)

**S1 Table. STROBE 2007 (v4) checklist of items to be included in reports of observational studies in epidemiology**[*]. Checklist for cohort, case-control, and cross-sectional studies (combined).
(DOC)

## Author Contributions

**Conceptualization:** Arjun K. Venkatesh, Alexander Janke, Robert D. Becher.

**Data curation:** Craig Rothenberg, Edwin Chan.

**Formal analysis:** Craig Rothenberg.

**Investigation:** Arjun K. Venkatesh, Alexander Janke, Craig Rothenberg.

**Methodology:** Arjun K. Venkatesh, Alexander Janke.

**Project administration:** Arjun K. Venkatesh, Alexander Janke, Robert D. Becher.

**Resources:** Craig Rothenberg, Robert D. Becher.

**Software:** Craig Rothenberg.

**Validation:** Craig Rothenberg.

**Visualization:** Alexander Janke, Craig Rothenberg.

**Writing – original draft:** Arjun K. Venkatesh, Alexander Janke.

**Writing – review & editing:** Arjun K. Venkatesh, Alexander Janke, Craig Rothenberg.

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
