## [Decision Letter · Decision Letter 0]

7 Jan 2021

PONE-D-20-35527

National Trends in Emergency Department Closures, Mergers and Utilization, 2005-2015

PLOS ONE

Dear Dr. Venkatesh,

Thank you for submitting your manuscript to PLOS ONE. After careful consideration, we feel that it has merit but does not fully meet PLOS ONE’s publication criteria as it currently stands. Therefore, we invite you to submit a revised version of the manuscript that addresses the points raised during the review process.

Please review and address reviewer and additional editor comments below.

We look forward to receiving your revised manuscript.

Kind regards,

Fernando A. Wilson, PhD

Academic Editor

PLOS ONE

Journal Requirements:

2. We note that Figure 1 and Supplemental Figure 1 in your submission contain map images which may be copyrighted.

a. You may seek permission from the original copyright holder of Figure 1 and Supplemental Figure 1 to publish the content specifically under the CC BY 4.0 license. 

Additional Editor Comments:

- Introduction:  write out EMTALA at first mention.  

    - delete comma after “competitive markets,”.

    - do you mean “recent national assessment of ED closures”?

- Follow PLOS ONE guidelines relating to naming of supplemental exhibits. Refer to:

https://journals.plos.org/plosone/s/submission-guidelines#loc-supporting-information

- Pg. 8:  - add “Our study shows” in first sentence.

            - Last paragraph:  add “in our analysis” after “hospital-based ED closures”.

- Pg. 10:  add “our findings suggest that there has been…” in 1st sentence.

- References must are formatted following PLOS ONE requirements.  Use journal abbreviations (eg Health Aff) and be consistent in use of capitalization in article titles.  The online articles are not correctly formatted.  Refer to the following link:

https://journals.plos.org/plosone/s/submission-guidelines#loc-references

- Tables need to be placed within the manuscript directly after the paragraph in which they are first mentioned. Figure captions must be inserted within the manuscript text immediately following the paragraph in which the figure is first mentioned.  Delete pages 14-15 listing the captions.

- Tables 1 and 2:  add commas for thousands.

- Figures 1a and 1b should either be formatted as a two-panel figure or relabeled as figures 1 and 2.  Also, the color of the ‘data suppressed’ regions is not consistent with that in the legend.

- Figures 2 and 3 have poor resolution and are difficult to read.  Please review the following guidelines on figures:

https://journals.plos.org/plosone/s/figures

Reviewers' comments:

Reviewer's Responses to Questions

**Comments to the Author**

1. Is the manuscript technically sound, and do the data support the conclusions?

Reviewer #1: Yes

Reviewer #2: Partly

2. Has the statistical analysis been performed appropriately and rigorously? 

Reviewer #1: Yes

Reviewer #2: Yes

3. Have the authors made all data underlying the findings in their manuscript fully available?

Reviewer #1: Yes

Reviewer #2: Yes

4. Is the manuscript presented in an intelligible fashion and written in standard English?

Reviewer #1: Yes

Reviewer #2: Yes

5. Review Comments to the Author

Reviewer #1: In this observational study, the authors utilize the American Hospital Association annual survey from 2005 to 2015 to describe trends in nationwide emergency department closures and mergers. The authors report 200 closures, 138 mergers, and 160 new emergency departments. While the overall number of emergency departments is relatively stable, the number of emergency department visits are increasing over time leading to current emergency departments having to manage larger patient volumes. This is a nice study however I do have several comments below:

1. Firstly I commend the authors on a thorough and tedious approach to identifying closures and mergers.

2. The way the RUCA codes were categorized is confusing to this reviewer. For example, RUCA codes 8.1 & 10.1 were defined as “urban” however per my review of the documentation provided by the USDA these codes represent small towns and rural areas respectively. Perhaps some further clarification and/or a ref to where this schema comes from would make this clearer.

3. Perhaps this reviewer missed this point but is it possible that an emergency department closes but a hospital remains open? In other words, are all of the ED closures captured part of a larger hospital closing? If it is possible to tease these out as being different could the authors comment on what it might mean for an ED to close but the hospital to remain open?

4. The authors note that due to data limitations they could not capture closures beyond 2014 however, two important trends have occurred since then. First many states have expanded Medicaid and secondly an increasing trend in hospital closures. Could the authors postulate what more recent ED trends may look like and their implications.

5. The second paragraph of the results sections states that closures and mergers were more likely for smaller EDs and those in urban areas. Furthermore, table 1 suggest that being rural is protective against closure (aORs 0.39, 0.27, & 0.51 respectively). I am surprised by this finding as most of the hospital closure literature suggest that rural hospitals are at greater risk of closure. Furthermore, lower ED volumes, which I would expect rural EDs to have, were associated with greater odds of closure. These two findings intuitively seem to contradict each other. Could the authors expand on these findings?

Reviewer #2: Major comments: This is a descriptive analysis of hospital-based emergency department closures that improves on prior work by also describing mergers and free-standing facilities. This work is particularly relevant given the current stress on acute care facilities due to the pandemic. The definition of closures has a high level of rigor, including review of news sources by research staff. The rationale is clear and the discussion is thoughtful, but I have a concerns about the statistical model that should be addressed. The analysis evaluates trends in ED/inpatient volume by rural status but not trends in ED closures. Evaluating closure trends by rural/urban status would improve the value and strengthen the conclusions of the article.

Minor comments:

What was the basis for requiring two consecutive years with >500 ED visits per year to define an ED? Consider giving a reference point for a typical rural ED to show this isn’t excluding low volume CAH or other rural hospitals.

I am concerned about potential collinearity of volume/rural status. What VIF was used as the threshold for multicollinearity and did the relationship of rural status to closure change depending on the inclusion of volume in the model?

Why did the model not control for year/time trends? Part of the rationale for the analysis is the trend towards consolidation, and it would be valuable to present year trends and test for significance of a year variable. If sufficient power, interact year with rural to compare time trends (collapsing rural categories may be necessary).

How was clustering of outcomes by year and HRR accounted for in the statistical analysis? consider a multi-level model.

Discussion “Our findings suggest that hospital-based EDs most likely to be closed or merged are low-volume facilities in urban areas.” I am not convinced the evidence presented supports concluding greater risk in urban areas – particularly recent trends. Based on the descriptive data presented in this analysis, the trends in ED closures appear to be increasing in rural areas and decreasing in urban areas. In 2013 and 2014, there were 26 urban closures and 34 rural closures (Table 1) -- guessing the denominator is smaller in rural, making the rate/risk even higher. The rural/urban trend analysis is needed to support this statement (see prior comments on statistical model).

Also, note fewer mergers in rural areas (Table 2) only 7 rural versus 24 urban (2013-2014) – again, not sure how to interpret without the denominators; consider discussing if/why merger trends differ in rural/urban?

The discussion briefly mentions the potential differential impact of closures for rural areas; this is an important caveat to the conclusion that access is stable and should be mentioned in the abstract/conclusion.

Table 1: Please also present an unadjusted OR for each variable. I found this table confusing, and it’s misleading to present counts without a denominator for each row. Ideally, I would reorganize the presentation with columns for urbanicity, and include year in the model so the aOR for year can be presented (see prior comment on methods to compare trends by urbanicity).

Figure 2: Consider instead presenting rates of mergers/closures by rural status over time.

Figure 3: the figure was blurry, couldn’t evaluate.

Supplemental figure 2: may be more valuable to present closures (by urbanicity) rather than all facilities (I would prefer this to the closure map in the main figures)

6. PLOS authors have the option to publish the peer review history of their article (what does this mean?). If published, this will include your full peer review and any attached files.

Reviewer #1: No

Reviewer #2: **Yes: **Brystana Kaufman

---

## [Author Response · Author response to Decision Letter 0]

11 Mar 2021

dditional Editor Comments:

- Introduction: write out EMTALA at first mention.

Done.

 - delete comma after “competitive markets,”.

Done.

 - do you mean “recent national assessment of ED closures”?

Yes, changed. 

- Follow PLOS ONE guidelines relating to naming of supplemental exhibits. Refer to:

https://journals.plos.org/plosone/s/submission-guidelines#loc-supporting-information

Done.

- Pg. 8: - add “Our study shows” in first sentence.

Done.

 - Last paragraph: add “in our analysis” after “hospital-based ED closures”.

Done.

- Pg. 10: add “our findings suggest that there has been…” in 1st sentence.

Done.

- References must are formatted following PLOS ONE requirements. Use journal abbreviations (eg Health Aff) and be consistent in use of capitalization in article titles. The online articles are not correctly formatted. Refer to the following link:

https://journals.plos.org/plosone/s/submission-guidelines#loc-references

Thank you, we have adjusted references to this requirement.

- Tables need to be placed within the manuscript directly after the paragraph in which they are first mentioned. Figure captions must be inserted within the manuscript text immediately following the paragraph in which the figure is first mentioned. Delete pages 14-15 listing the captions.

Done.

- Tables 1 and 2: add commas for thousands.

Done.

- Figures 1a and 1b should either be formatted as a two-panel figure or relabeled as figures 1 and 2. Also, the color of the ‘data suppressed’ regions is not consistent with that in the legend.

Figure 1 has been combined into a panel. To clarify the ‘data suppressed’ regions: the hospital referral regions do not actually cover all area in the continental United States, so some areas are ‘data suppressed’ to meet the data use agreement for the American Hospital Association Annual Survey, while other areas may appear white as they are not contained within any hospital referral region. We confirmed that this is the standard manner of mapping hospital referral regions (see CMS release on hospital charge variation here*). We added the following sentence to better clarify in the caption to Figure 1:

“Note that HRRs do not cover all land in the continent U.S., these areas on the map will not be shaded.”

*CMS releases annual update to data on hospital charge variation. Accessed January 25, 2021. URL: https://www.cms.gov/newsroom/fact-sheets/cms-releases-annual-update-data-hospital-charge-variation

All maps used in our manuscript were created using R software (version 3.6.1) based on un-copyrighted, publicly available map projections as source data.

- Figures 2 and 3 have poor resolution and are difficult to read. Please review the following guidelines on figures:

https://journals.plos.org/plosone/s/figures

Figures 2 and 3 have been updated/modified to improve readability.

 

Reviewers' comments:

Reviewer's Responses to Questions

Comments to the Author

1. Is the manuscript technically sound, and do the data support the conclusions?

Reviewer #1: Yes

Reviewer #2: Partly

2. Has the statistical analysis been performed appropriately and rigorously?

Reviewer #1: Yes

Reviewer #2: Yes

3. Have the authors made all data underlying the findings in their manuscript fully available?

Reviewer #1: Yes

Reviewer #2: Yes

4. Is the manuscript presented in an intelligible fashion and written in standard English?

Reviewer #1: Yes

Reviewer #2: Yes

5. Review Comments to the Author

 

Reviewer #1: In this observational study, the authors utilize the American Hospital Association annual survey from 2005 to 2015 to describe trends in nationwide emergency department closures and mergers. The authors report 200 closures, 138 mergers, and 160 new emergency departments. While the overall number of emergency departments is relatively stable, the number of emergency department visits are increasing over time leading to current emergency departments having to manage larger patient volumes. This is a nice study however I do have several comments below:

1. Firstly I commend the authors on a thorough and tedious approach to identifying closures and mergers.

2. The way the RUCA codes were categorized is confusing to this reviewer. For example, RUCA codes 8.1 & 10.1 were defined as “urban” however per my review of the documentation provided by the USDA these codes represent small towns and rural areas respectively. Perhaps some further clarification and/or a ref to where this schema comes from would make this clearer.

This is an excellent point that should be clarified. Our citation from the U.S.D.A. Economic Research Service [23] explains the RUCA codes, while our citation from the Rural Health Research Center [24] explains why the tracts are grouped as they are for our research study as in other healthcare-related analyses. We have added additional language to the methods section to clarify why these codes are distributed in this manner:

“These groupings are standardized from the Rural Health Research Center, and sort tracts according to patterns of commute (for example, a rural tract with 30-50% of commuting secondary flow to an urban area is categorized as urban) [23].”

3. Perhaps this reviewer missed this point but is it possible that an emergency department closes but a hospital remains open? In other words, are all of the ED closures captured part of a larger hospital closing? If it is possible to tease these out as being different could the authors comment on what it might mean for an ED to close but the hospital to remain open?

Yes, it is possible that an ED closed but that the associated hospital remained open. Our study defines ED closure either according to a fall in yearly ED visit count below 500 or if the hospital closed. We added the following phrase to the Methods, Outcomes to clarify:

“Some hospitals may remain open while their associated ED is closed, and these fit our outcome definition for a hospital-based ED closure.”

4. The authors note that due to data limitations they could not capture closures beyond 2014 however, two important trends have occurred since then. First many states have expanded Medicaid and secondly an increasing trend in hospital closures. Could the authors postulate what more recent ED trends may look like and their implications.

This is an excellent point. Although we do not have data to specifically address how the incidence of closures and mergers evolved after 2014, we have adjusted this language in the discussion to more directly address this issue:

“This may be attenuated by the Affordable Care Act in 2010, and subsequent to statewide Medicaid expansions and other shifts in underlying population insurance status beyond 2014, hospital-based EDs may be less prone to closure.”

5. The second paragraph of the results sections states that closures and mergers were more likely for smaller EDs and those in urban areas. Furthermore, table 1 suggest that being rural is protective against closure (aORs 0.39, 0.27, & 0.51 respectively). I am surprised by this finding as most of the hospital closure literature suggest that rural hospitals are at greater risk of closure. Furthermore, lower ED volumes, which I would expect rural EDs to have, were associated with greater odds of closure. These two findings intuitively seem to contradict each other. Could the authors expand on these findings?

This was an interesting finding, and we’ve provided some further language in the discussion section around hospital consolidation on why this might be the case:

“This phenomena may also explain our finding that small EDs in urban areas are more likely to close, as opposed to rural sites, as they may be more likely to be subject to competitive pressures while small rural EDs that are more commonly threatened with closure in media reports may be able to raise political or community attention about the essential role of rural EDs in providing access to complex care not available elsewhere.”

 

Reviewer #2: Major comments: This is a descriptive analysis of hospital-based emergency department closures that improves on prior work by also describing mergers and free-standing facilities. This work is particularly relevant given the current stress on acute care facilities due to the pandemic. The definition of closures has a high level of rigor, including review of news sources by research staff. The rationale is clear and the discussion is thoughtful, but I have a concerns about the statistical model that should be addressed. The analysis evaluates trends in ED/inpatient volume by rural status but not trends in ED closures. Evaluating closure trends by rural/urban status would improve the value and strengthen the conclusions of the article.

Minor comments:

What was the basis for requiring two consecutive years with >500 ED visits per year to define an ED? Consider giving a reference point for a typical rural ED to show this isn’t excluding low volume CAH or other rural hospitals.

This is an important question. The AHA data, composed of individual hospitals’ survey responses, may contain sites where an ED is reported, but that do not actually represent a meaningful point of emergency care services. To avoid categorizing such sites as EDs, we set this a standard within the data. Reassuringly, we identified an overall number of hospital-based EDs consistent with prior publications now reported in the results, and we added the following language to better clarify this reasoning and make a clarification about the order of magnitude of volume for rural EDs [21] now reported in the methods. 

“As a point of comparison, our approach identified 4,460 EDs in 2015, similar to the 4,545 EDs in the same year according to data from the Healthcare Cost and Utilization Project [26].”

“This approach, requiring 2 years of consecutive data and >500 ED visits per year, was set to exclude potentially spurious sites in this dataset composed of aggregated hospital survey responses. As a point of reference, there was an estimated 28.4 million ED visits and 1,855 rural EDs in 2016, so that the average yearly ED visits per rural ED would be 15,309 [21], while sites with <500 ED visits per year were felt to be less likely to represent a bona fide point of emergency care services.”

I am concerned about potential collinearity of volume/rural status. What VIF was used as the threshold for multicollinearity and did the relationship of rural status to closure change depending on the inclusion of volume in the model?

This is an important point. We obtained associated variance inflation factors to assess for multicollinearity amongst independent variables. We used a cutoff of below three, and in all cases met this standard.

Why did the model not control for year/time trends? Part of the rationale for the analysis is the trend towards consolidation, and it would be valuable to present year trends and test for significance of a year variable. If sufficient power, interact year with rural to compare time trends (collapsing rural categories may be necessary).

We agree with the reviewers sentiment that adjusting for time within the model would better enable us to make statistical inferences about the rate of closure or merger. However, given the fairly rare rate of occurrence of the outcomes and our focus on descriptive characterization of closures and mergers we have ensured to avoid language that may be misinterpreted as a statistical statement on the rate of change. Instead we believe our current descriptive results and regression outcomes are better aligned with a statistical model without time adjustment, particular given that most covariates in the model are fairly statistic over time without sufficient underlying variation for time to be a meaningful variable. We included the following language in the methods section to make this clear to the reader:

“Our analysis does not include time trends, as event rates for closure and merger were fairly uncommon to derive stable estimates, and variation in other ED characteristics generally static over time, to provide a definitive analysis of how factors associated with closure and merger have changed over time..”

How was clustering of outcomes by year and HRR accounted for in the statistical analysis? Consider a multi-level model.

We appreciate the importance of accounting for clustering. However the only analysis conducted at the HRR level are those used for descriptive statistics depicted in Figure 1 as to avoid the identification of hospitals per the Data Use Agreement with the AHA. In contrast, all regression analyses are conducted at the hospital level without any clustering of observations that would necessitate a mixed-effects or hierarchical model.

Discussion “Our findings suggest that hospital-based EDs most likely to be closed or merged are low-volume facilities in urban areas.” I am not convinced the evidence presented supports concluding greater risk in urban areas – particularly recent trends. Based on the descriptive data presented in this analysis, the trends in ED closures appear to be increasing in rural areas and decreasing in urban areas. In 2013 and 2014, there were 26 urban closures and 34 rural closures (Table 1) -- guessing the denominator is smaller in rural, making the rate/risk even higher. The rural/urban trend analysis is needed to support this statement (see prior comments on statistical model).

That our results do not distinguish between recent trends and overall picture in the full time period of analysis is an notable limitation. We have added the following language to clarify that these results are a statement of the ‘static’ risk across the full sample rather than a statement about how the risk of closure or merger has changed over time.

“Our findings suggest that, throughout the full analysis period, hospital-based EDs most likely to be closed or merged are low-volume facilities in urban areas.”

Also, note fewer mergers in rural areas (Table 2) only 7 rural versus 24 urban (2013-2014) – again, not sure how to interpret without the denominators; consider discussing if/why merger trends differ in rural/urban?

This is a good point, and we added language as above to clarify that we aim to avoid making strong statements about the trend in rates of closure, merger, and factors associated with those outcomes, given the low yearly event rates. To better clarify this, we added the following to the results section and this includes a statement about the ‘denominators’ in this case.

“In the final year of analysis, there were 2,296 urban hospital-based EDs and 2,164 large rural, small rural, or isolated hospital-based EDs. The event rates for closure and merger, as a percent of overall EDs, remained <1% across subgroups of EDs and across years.”

The discussion briefly mentions the potential differential impact of closures for rural areas; this is an important caveat to the conclusion that access is stable and should be mentioned in the abstract/conclusion.

We intend to highlight this point and we’ve made the following changes to the abstract and the conclusion, respectively, to better address this:

“The number of hospital-based ED closures is small when accounting for mergers, but occurs as many more patients are presenting to a stable number of EDs in larger health systems, though rural areas may be differentially affected. EDs were managing accelerating patient volumes alongside stagnant inpatient bed capacity.”

“Though hospital-based EDs are closing and consolidating, our findings suggest that there has been very little change in the overall number of facilities nationwide, with more patients presenting to a stable number of emergency departments in larger, more complex systems, though rural areas may be differentially affected.”

Table 1: Please also present an unadjusted OR for each variable. I found this table confusing, and it’s misleading to present counts without a denominator for each row. Ideally, I would reorgaasnize the presentation with columns for urbanicity, and include year in the model so the aOR for year can be presented (see prior comment on methods to compare trends by urbanicity).

Unadjusted odds ratios are now reported in Tables 1 and 2. We agree with the reviewers sentiment that adjusting for time within the model would better enable us to make statistical inferences about the rate of closure or merger. However, given the fairly rare occurrence of the outcomes and our focus on descriptive characterization of closures and mergers we have ensured to avoid language that may be misinterpreted as a statistical statement on the rate of change. Instead we believe our current descriptive results and regression outcomes are better aligned with a statistical model without time adjustment, particularly given that most covariates in the model are fairly static over time without sufficient underlying variation for time to be a meaningful variable.

Figure 2: Consider instead presenting rates of mergers/closures by rural status over time.

Given the rare occurrence of mergers and closures as a percent of overall EDs, we had preferred to avoid reporting rates. We added the language to the results section to clarify how rare these events were by this measure.

“The event rates for closure and merger, as a percent of overall EDs, remained <1% across subgroups of EDs and across years.”

Figure 3: the figure was blurry, couldn’t evaluate.

Figures 2 and 3 have been updated/modified to improve readability.

Supplemental figure 2: may be more valuable to present closures (by urbanicity) rather than all facilities (I would prefer this to the closure map in the main figures)

You have brought to our attention an important limitation resulting from the data use agreement. Just as we must report our results by hospital referral region in Figure 1 due to the DUA to avoid identifying hospital, we have to remove Supplemental Figure 2 for the same reason. This issue is mitigated partially given the urban-rural distinctions can be clarified via citations in the text and are relatively intuitive.

---

## [Editor Report · Decision Letter 1]

14 Apr 2021

PONE-D-20-35527R1

National Trends in Emergency Department Closures, Mergers and Utilization, 2005-2015

PLOS ONE

Dear Dr. Venkatesh,

Thank you for submitting your manuscript to PLOS ONE. After careful consideration, we feel that it has merit but does not fully meet PLOS ONE’s publication criteria as it currently stands. Therefore, we invite you to submit a revised version of the manuscript that addresses the points raised during the review process.

Specifically, the "clean" version of the manuscript without track changes seems to be the original submission. Please remove the old manuscript file and upload the revised version together with the track changes version of the manuscript.  Please refer to: https://journals.plos.org/plosone/s/revising-your-manuscript

We look forward to receiving your revised manuscript.

Kind regards,

Fernando A. Wilson, PhD

Academic Editor

PLOS ONE

Journal Requirements:

Additional Editor Comments (if provided):

- The "clean" version of the manuscript without track changes seems to be the original submission.  Please remove the old manuscript file and upload the revised version together with the track changes version of the manuscript.  Please refer to: https://journals.plos.org/plosone/s/revising-your-manuscript

---

## [Author Response · Author response to Decision Letter 1]

14 Apr 2021

1) Specifically, the "clean" version of the manuscript without track changes seems to be the original submission. Please remove the old manuscript file and upload the revised version together with the track changes version of the manuscript. 

We have remove the old version and uploaded the clean version of the manuscript as requested.

---

## [Editor Report · Decision Letter 2]

3 May 2021

National Trends in Emergency Department Closures, Mergers and Utilization, 2005-2015

PONE-D-20-35527R2

Dear Dr. Venkatesh,

We’re pleased to inform you that your manuscript has been judged scientifically suitable for publication and will be formally accepted for publication once it meets all outstanding technical requirements.

Kind regards,

Fernando A. Wilson, PhD

Academic Editor

PLOS ONE
---

## [Editor Report · Acceptance letter]

6 May 2021

PONE-D-20-35527R2 

National trends in emergency department closures, mergers, and utilization, 2005-2015 

Dear Dr. Venkatesh:

I'm pleased to inform you that your manuscript has been deemed suitable for publication in PLOS ONE. Congratulations! Your manuscript is now with our production department. 

Kind regards, 

on behalf of

Dr. Fernando A. Wilson 

Academic Editor

PLOS ONE